# Perceptions of Mothers about Support and Self-Efficacy in Breastfeeding: A Qualitative Study

**DOI:** 10.3390/children9121920

**Published:** 2022-12-08

**Authors:** Esther Gálvez-Adalia, Raquel Bartolomé-Gutiérrez, Carlos Berlanga-Macías, Beatriz Rodríguez-Martín, Irene Marcilla-Toribio, María Martínez-Andrés

**Affiliations:** 1Health Service of Castilla-La Mancha, SESCAM, 16071 Cuenca, Spain; 2Faculty of Nursing, Universidad de Castilla-La Mancha, 02071 Albacete, Spain; 3Health and Social Research Center, Universidad de Castilla-La Mancha, 16071 Cuenca, Spain; 4Faculty of Health Sciences, Universidad de Castilla-La Mancha, 45600 Talavera de la Reina, Spain; 5Health, Gender and Social Determinants Research Group, Universidad de Castilla-La Mancha, 16071 Cuenca, Spain

**Keywords:** breastfeeding support, exclusive breastfeeding, self-efficacy, health knowledge, maternal behaviour, qualitative study

## Abstract

Breastfeeding is a complex process influenced by different personal and social factors which will determine both the initiation and the resilience for its maintenance. The aim is to identify the beliefs and expectations of mothers concerning breastfeeding to determine the perception of their self-efficacy and the influence on the management of their babies’ feeding. A qualitative study through semi-structured interviews was carried out. The sample size was defined by the saturation criteria. Twenty-two women participated, eleven were from an urban environment and eleven were from a rural environment. Mothers’ knowledge of breastfeeding, their expectations of that process, their experience, and their strategies for overcoming problems associated with initiating, establishing, and continuing breastfeeding were influenced by the role of nurses and midwives in supporting their perception of self-efficacy. Likewise, maternity policies are important for the continuance of exclusive breastfeeding. This study shows the complexity of the initiation and establishment of breastfeeding and the existence of several social factors surrounding these moments. Furthermore, it demonstrates the importance and reference of nurses and midwives and the role of State maternity policies.

## 1. Introduction

The scientific literature evinces the benefits of breastfeeding for the health of both mothers and their babies [1]. Particularly important for the development of the newborn is the influence that the composition of breast milk has, not only on physical growth but likewise on the reduction of infectious diseases as well as childhood obesity, and even encouraging a better cognitive performance [2,3,4]. For the mother, from the influence of suckling the infant on the production of oxytocin, which facilitates better uterine involution, to the reduction in ovarian cancer risk or even postpartum depression, there are an entire list of benefits for the physical and psychological health of women [5].

Prestigious international organisations such as the World Health Organization (WHO) and the United Nations International Children’s Emergency Fund (UNICEF) insist on the benefits of establishing exclusive breastfeeding for the first six months and its complementary continuation for at least two years. Nevertheless, we have come across higher than desirable rates of abandonment [6,7,8]. 

A woman’s decision to breastfeed is a process influenced by a myriad of factors which will determine both the initiation of breastfeeding, and the resilience allows for its maintenance in the face of adversities which may come about in a process that is generally idealized [9]. Despite being a personal decision, the woman will draw on all sources of experience and role models from everyone who is around her in order to choose one option or another [10]. Accordingly, finding that the decision may be influenced by personal factors such as previous breastfeeding experiences, having more than one child, partner support; structural factors such as access to maternal education and midwifery services; and social factors such as access to informal sources of information via the internet [11].

In addition to this, women find it difficult to reconcile professional and family life, with work being one of the reasons for abandoning breastfeeding [12]. Thus, efforts to maintain their occupational status becomes an added pressure regarding their decision whether to continue long-term breastfeeding or not.

Expectations created based on these previous experiences and the information received can lead to a conflict with the reality of the situation. Feelings of guilt are thereby generated, whereby the woman perceives that she is judged by her environment, and even judges herself by assessing that she is “not being a good mother”. The very act of breastfeeding itself becomes a guilt-inducing trigger which grows and evolves according to the mother’s vulnerability and the pressure from the social environment concerning maternal decisions [13]. A negative perception of maternal self-efficacy may lead to early weaning. The belief of an inability to control the difficulties is propitious to the abandonment of the task by minimising effort [14].

There is therefore a need for maternal support which assists the creation of a positive environment to foster initiating and the continuance of exclusive breastfeeding. Breastfeeding women hope to receive this support before the birth not only in the informal setting from their peers, but likewise from health care professionals. Pregnant women attach importance to that feeling that someone is listening and to confide in that person who will be there to provide appropriate support through information which strengthens the relationship with that professional [15,16].

To date, studies on breastfeeding are mainly based on the benefits of breastfeeding and those that study maternal perceptions for the most part quantify the duration of breastfeeding and associated factors which affect the breastfeeding mother, none of the qualitative studies were undertaken in Spain [12,15,16,17,18,19,20,21,22]. In these studies, the researchers showed breastfeeding rates at six months, between 15.5% and 25.4%, and factors to stopping breastfeeding, such as low milk supply or work-related causes. There is a need for approaches to be undertaken considering maternal experiences and perceptions which enables the identification of the reasons as to why mothers make decisions regarding breastfeeding and its continuance. Thus, understanding mothers may carry out more effective interventions to support and maintain breastfeeding [12,15,16,17,18,19].

## 2. Materials and Methods

### 2.1. Objective

To identify the beliefs and expectations of mothers concerning breastfeeding to determine the perception of their self-efficacy and the influence on the management of their babies’ feeding.

### 2.2. Methods

The methodology has focused on the theoretical framework provided by the Social Cognitive Theory (SCT) [10]. This theory enables to human behaviour to be explained by taking into account the influence of the social imaginary, the realities encountered, and the social and personal expectations which will factor in the development of the self-concept and decision-making, as well as in the conflict resolution styles that one undertakes. 

The SCT of gender role development and functioning is the ideal ground for understanding breastfeeding and the processes surrounding the matter, conditioned by gender-regulated behaviours through affirmations and perceptions of self-efficacy learned through experiences and models [23]. Due to this, it is pertinent to address our research with a gendered perspective, in order to explain women’s lived experiences, in a particular setting and with a specific social and cultural history.

In this regard, the qualitative design following the SCT enables addressing women’s experiences through the subject analysis of semi-structured interviews [24,25] in order to ascertain and assess the expectations, beliefs, knowledge, and lived realities of breastfeeding mothers and their interrelation with personal, social, and cultural characteristics [26]. Furthermore, the need for so-called cultural competence in healthcare personnel, mainly nurses, renders qualitative research a fundamental resource for how to provide the necessary health support, or for how to improve the care supplied [27]. 

### 2.3. Sample

This qualitative research is part of the MOVI-da 10! [28] study investigating the influence on cardiovascular health and cognitive performance of recreational and non-competitive physical activity intervention in school children (4–6 years old), with the participation of 8 schools in the province of Cuenca, Spain. Part of this study assessed the type and duration of breastfeeding received, proposed to mothers to participate in the nested study as regards their motherhood. A purposive sampling [29] was undertaken among those women who showed interest in participating. The research protocol, which was developed for the qualitative study, was published in April 2021 [30]. 

The qualitative study participants are women residents in the province of Cuenca, Spain, who had a normal pregnancy and had given birth between six months and three years prior to their collaboration with us, in order to eschew wide variations in the attention to the process, as well as possible memory bias. 

Finally, 22 women participated, a sample size defined by the saturation of information criteria [29]. Of these, 11 were from an urban environment and 11 from a rural environment, belonging to different levels of education. From among the rural women, 3 were immigrants, 1 of Arab origin and 2 of Latin American origin.

### 2.4. Location

The interviews were conducted in person by a single interviewer (M.MA). The possibility of holding interviews in several locations was offered in order to create a pleasant atmosphere which would facilitate the verbalisation of their experiences [29]. These locations were distributed according to the women’s area of residence; in reference to the urban area in the research centre itself, their homes or workplaces, and, as regards the rural area, their homes or the reference health centre. 

### 2.5. Data Collection

The semi-structured interviews were conducted on the basis of a preliminary script, with the most significant aspects defined in the previous documentation work and as per the SCT model. Furthermore, two pilot interviews were conducted to assess the suitability thereof [31]. These were conducted between March and June 2018. The script focuses on three main sections: (i) Motherhood, (ii) Pregnancy, Childbirth, and Postpartum and (iii) Breastfeeding. In the present study, our analysis has focused on the breastfeeding section (Table 1).

To ensure quality, face-to-face interviews were conducted by the same researcher, with experience in qualitative methodology and communication skills, establishing a relationship of trust that facilitated the free dialogue and communication of the woman to verbalise their expectations and experiences, and were recorded for later transcription and analysis [25]. 

The interview period concluded following the saturation of information criteria. The interviews were transcribed using the F4 software, and from a first reading of the interviews the most relevant ideas were educed, which were observed in a general manner in all of these. Thus, an overall summary of such ideas was undertaken, which was forwarded to the participating mothers so that they could assess whether this summary responded to the considerations which these women proffered in their various interviews. None of the participants related any clarifications, nor mentioned that they did not feel represented in the ideas disclosed.

### 2.6. Data Analysis

In relation to the subsequent data analysis, the ATLAS TI 8.3 software was used. This analysis was performed by triangulating the data with 3 experienced researchers (E.G.-A., M.M.-A. and R.B.-G.), who used conventional content analysis to delve deeper into the text. The coding of the texts was undertaken at three levels: code, category, and issues [32,33]. The process will thereby allow the information to be organized by groups and integrated according to the theoretical SCT model.

### 2.7. Ethical Aspects

This study has the approval of the Clinical Research Ethics Committee of the Hospital Virgen de la Luz of Cuenca, obtained for the MOVI-da 10! intervention of which it forms a part (approval number: 2016/PI0216; approval date: 11 October 2016). In addition to informed consent, the participants were advised at the beginning of the interviews as regards the recording of the interviews and the assignment of an identification code in order to anonymise the interviews. The participants were likewise reminded that they could revoke their consent during any part of the research process.

### 2.8. Validity and Meticulousness

Several strategies have enabled us to ensure a meticulous analytical approach. The sampling technique used enhances the reliability and credibility of the findings by affording access to a heterogeneous population, albeit with a shared cultural reality (motherhood in Western society).

The settings selected were sufficiently comfortable to enable the women to express themselves freely. Validity was reinforced by means of the systematic use of semi-structured interview techniques conducted by an experienced researcher. Furthermore, the forwarding of a summary of the information obtained to the participants permitted these women to review the content in order to validate or reject anything in the event of disagreement. Finally, analysis and coding with three different researchers at three degrees of depth allowed for the subsequent discussion of possible discrepancies and the resolution thereof via consensus [33].

## 3. Results

### 3.1. Sample Characteristics

Twenty-two females participated in the study. Eleven females (50%) belonged to a rural environment and the remaining 50% from an urban one. Among rural women, three of them were immigrants, one of Arab origin and two of Latin American origin. The socio-demographic data obtained are shown in Table 2.

### 3.2. Breastfeeding Knowledge

The participants were aware of the benefits of breastfeeding for both their children as well as themselves. Nevertheless, the maternal benefits were the least discussed, focusing on the health of their baby. With the exception of one of the women, all of them considered exclusive and as-needed breastfeeding as their first choice and their intention to continue breastfeeding as long as possible, even attaching importance to breastfeeding in the mother-baby relationship.


*“The midwife told us to do it if we could as it was the best and the most natural thing to do and then you try it… if you can do it, why not?” ID8*



*“You’ve always been told that it’s the best/the best food for your baby” ID66*



*“because of the immune system/of course I liked it… we already have a bond” ID25*


The participants consider breastfeeding as a personal choice of the mothers. Despite this, the participants are critical of those who do not have breastfeeding as their first choice. It is only perceived as justifiable when having trouble breastfeeding the baby.


*“I think everyone is free to do as they see fit. But it also depends, there are women who can’t breastfeed. So, woman, if you can’t, you can’t.” ID33*



*“It’s a very personal issue/I find it strange that a mother doesn’t want to breastfeed from the beginning/because I thought it was such a great thing for my children… it’s difficult for me to understand them” ID16*


### 3.3. Expectations and Reality of Breastfeeding

Mothers who breastfed considered that breastfeeding would be their first choice of feeding for their baby from the very beginning.


*“You leave there [the antenatal classes] as if to say I’m going to breastfeed no matter what” ID48*



*“Yes/I wanted to try/I guess that this happens to everyone…” ID13*



*“If I could and as long as I could I wanted to breastfeed” ID62*


Some of the mothers imagined a simpler process, that breastfeeding would be established from the very first moment. The mothers did not consider breastfeeding as a learning process. To the extent that problems arose, some mothers considered that they had not been prepared for that and leading to the surfacing of negative feelings.


*“It was not explained to me how hard breastfeeding is//because it is very hard, especially at the beginning until it is established” ID48*



*“There are moments when you are so sensitive and to be told certain things is like sinking into depression, a hopelessness, it’s like saying what am I doing wrong?” ID65*



*“To know if you have a good breast or good milk you have to have a bit of patience, as I say” ID46*



*“And they say “Do you breastfeed?” And I would say “no, I don’t, I bottle feed.” And it’s like “so you bottle feed the baby”, I say yes, “ah because you don’t have any breast milk, you can’t breastfeed”” ID26*


On certain occasions, breastfeeding was complicated by problems such as mastitis or the belief that the baby was not properly fed. This led to feelings of failure, guilt, frustration, and even depression if these problems did not allow the continuance of breastfeeding.


*“For a mother to be told that it’s not worth the effort to breastfeed as I couldn’t or that it’s not good milk… I wanted to forget everything” ID30*



*“Whatever happens/the problem is like in my case when you want to and you can’t/then there is a train wreck eh/because hormonally I felt very weak” ID48*



*“I feel bad to be honest. It’s like when other mothers talk about breastfeeding and such. It makes me feel bad to say that I haven’t been able to breastfeed.”ID28*



*“I had a hard time in that regard. Because I felt that I couldn’t feed my son”ID26*



*“I felt… that something could be happening to my girl and wouldn’t say anything. And I say, “what if I get dehydrated or something”” ID35*


### 3.4. Role of Healthcare Professionals

Midwives and nurses are the reference figures for the participants. The participants considering the role of these professionals to be especially important for the successful initiation and continuance of breastfeeding, although several of the latter felt very pressured by the former to continue breastfeeding.


*“I think it is fundamental for me that a person dressed in white as a healthcare worker in whom I place all my trust” ID48*



*“I burst into tears and I’m not prone to bursting into tears and I broke down and cried and [the paediatrician] immediately checked to see what was wrong with me. She called the midwife, my midwife, as the midwife was already talking to me, she was teaching me how to breastfeed” ID8*



*“If a nurse had come, well, look, place the baby onto the breast like this, see, I don’t think your milk has come in, let’s try… I’m going to place a breast pump on you, let’s help you like this/I don’t know/any advice that any of those women dressed in white would have given me, I would have set it in stone” ID48*



*“Then they also instil the fear into you that if you bottle feed the babies will not want the breast when breastfeeding” ID33*



*“If one of them had come and taught me or told me or explained how to place my daughter onto the breast, I wouldn’t have gone through half the woes that I experienced” ID26*


This pressure even leads mothers to lie in order not to be judged by nurses or midwives for not undertaking exclusive breastfeeding.


*“It depends on the nurse assigned to you. My eldest daughter was bottle-fed too. Back then everything was very controlled, ‘How many bottles a day do you give her? I would tell her a lesser number than that which I was giving my daughter, so that she could see that I was weaning the baby off the bottle, but that was a lie, I wasn’t weaning anything off.” ID35*



*“The conclusion that you draw from the preparation classes is that everyone can breastfeed/that it is the best thing in the world// […] the problem is, as in my case, when you want to and you can’t” ID48*



*“There are some moments/when you have just given birth, there are certain moments in your head that if you don’t get good advice, you are so lost and you don’t know if you are doing it right” ID13*


### 3.5. Institutional Support

The participants were very critical of the lack of support for continuing exclusive breastfeeding beyond statutory maternity leave. This was in addition to the length of maternity leave offered by the type of contract held by the participants, whether as an employee or self-employed worker.


*“I knew that if I could and as long as I could, I wanted to breastfeed. I was also clear about one thing, that once I started working I would not be able to do both. Namely, I wasn’t going to be breastfeeding, then bottle-feeding and so on. That when I started working I would stop.” ID14*



*“I think that women are not excessively given help in comparison to other countries” ID22*


Which is why the participants consider it necessary to increase the length of maternity leave in order to improve their chances of continuing exclusive breastfeeding.


*“If you can breastfeed a baby up to six months, well then you can introduce food and so on, but maybe yes, perhaps more people would try extra with their child on the breast and such. But of course, if your four months of maternity leave has finished and you have to return to work or you have to bottle feed the baby or you prepare the milk using a breast pump or whatever you do.” ID62*



*“Most women work, so if you take maternity leave to look after the baby for four months, five months maternity leave at the most, you have to return to work, if you don’t want to lose your job” ID8*



*“I think at least, a minimum of six months. I know it’s an awful lot, but at least six months” ID16*



*“I can consider myself lucky, I had my stipulated 16 week maternity leave and so on, but what about the self-employed women?” ID22*


## 4. Discussion

The purpose of our study was to identify the beliefs and expectations of mothers concerning breastfeeding to determine the perception of their self-efficacy and the influence on the management of their babies’ feeding. The aspects which were assessed were related to their knowledge of breastfeeding, their expectations of the breastfeeding process and their experience, and their strategies for overcoming problems associated with initiating, establishing, and continuing breastfeeding. These aspects were significantly related with the role of nurses and midwives in supporting their perception of self-efficacy. Likewise, maternity policies are important for the continuance of exclusive breastfeeding.

Our findings show that mothers are aware of the benefits of breastfeeding, and, despite this knowledge, it is not an associated factor for the continuance of breastfeeding. These findings are contrary to previous studies and systematic reviews demonstrating the relationship between the mothers’ received education as regards breastfeeding and the initiation and continuance thereof [34,35]. 

No matter which option and duration of breastfeeding is chosen by the participating mothers, all of them considered that the only justification for not providing exclusive breastfeeding were problems in the establishment thereof. These findings point to those obtained by Hunt et al. [36] in what is called the mechanism an/cannot conception of breastfeeding. It is defined to the author as “mechanistic constructions of breastfeeding and a rules-based approach contributed towards women polarising themselves as those who could or could not breastfeed and did or did not comply with the ‘rules’ for ‘successful’ breastfeeding” [36]. This mechanism is used both socially as well as medically [36,37,38]. Thus, women who are not able to carry out the technique of breastfeeding suffer from feelings of frustration and guilt, which may be related to depression and anxiety [38].

The participants consider nurses and midwives to be their basic reference for the initiation and establishment of breastfeeding, in line with other studies [18,39,40]. In the first moments after birth, mothers turn to the nurses and midwives for guidance on problems and doubts regarding to breastfeeding [18]. In the case of first-time mothers, this reference is considered essential even in those who have had no problems in establishing breastfeeding during the first hours of the baby’s life. Social support remains in the background and does not follow the findings of previous studies that point to social support, particularly in groups of other mothers and grandmothers, as a factor associated with the continuance of breastfeeding [10,12,40]. 

This figure of the nurse and midwife as a point of reference and support is likewise perceived as a pressure figure. For the participants, this fact entails a rejection and an erosion of the created mother-nurse relationship and is in line with the aforementioned an/cannot conception of breastfeeding mechanism [36,37]. This mechanism leads to mothers withholding valuable information from nurses as regards their breastfeeding problems that could otherwise entail professionals adapting their advice and assist in attaining a return to exclusive breastfeeding. It likewise sets a precedent which goes beyond breastfeeding and does not create a relationship of trust for future communication [19].

Together with other articles, the mothers in the study considered their incorporation into the workforce to be a fundamental difficulty in maintaining breastfeeding [12,17,18,41,42,43]. In the case of Spain, maternity leave is 16 weeks, and our participants evinced the requirement to increase this duration to at least six months to facilitate the continuance of breastfeeding. This recommendation to support breastfeeding by governments is underpinned by organisations such as UNICEF or the International Labour Organization (ILO) [41,44], and several studies have shown that extended maternity leave increases the duration of breastfeeding. 

Nevertheless, our participants expressed doubts as regards this relationship between maternity leave and duration of breastfeeding depending on the type of work, as they consider that mothers who are self-employed will not take maternity leave for work-related reasons. In Spain in 2022, women affiliated to the self-employed workers’ regime accounted for 36.2% of the 2,027,994 total [45], a percentage which has been gradually increasing [46]. This is why public policies for the protection of mothers should likewise include specific measures for self-employed women. 

### Strengths and Limitations

The organisation of the data collection was complicated by mainly environmental and social factors. Encountered, among these, was the difficulty for the mothers to spare the time necessary for conducting the interviews, whether due to their work commitments outside the home, or due to the burden of caring for their children incumbent upon these women, or the difficulty of travelling. Even so, participation was facilitated as far as was feasible by offering various locations and times that were more favourable for the women.

Data collection via semi-structured interviewing could be skewed by the opinion of the researcher but was considered appropriate given the intimate nature of the issues. Furthermore, several strategies sought to avoid these problems, such as recording and transcriptions of the interviews, cross-checking information with the participants themselves, or the triangulation in the data coding and analysis [26].

Although undertaking the study in a specific setting such as the province of Cuenca may preclude the generalisation of the data, the purposive sampling strategy has ensured having a heterogeneous sample in terms of experiences and urban/rural settings. In this fashion, the representativeness of the sample has been improved so that our results can be related to the Western female population of reproductive age [47].

## 5. Conclusions

The findings of this research show the complexity that exists at the initiation and establishment of breastfeeding, and the existence of several social factors surrounding these moments. It is worth highlighting the widespread knowledge of the importance of breastfeeding. In Spain in 2022, women affiliated to the self-employed workers’ regime accounted for 36.2% of the 2,027,994 total among women, as well as the an/cannot conception of breastfeeding mechanism at work. Furthermore, it demonstrates the importance and reference of nurses and midwives in the process both positively and negatively, and the role of State maternity policies.

Our findings suggest that efforts should be made at both the health and policy levels. In the former, it would be advisable to pursue breastfeeding support frameworks that go beyond the logic of the biomedical model and a foster woman-centred model which considers the socio-cultural reality surrounding infant feeding. Thus, future research efforts should focus on undertaking interventions about breastfeeding education programs addressed to health professionals and mothers during and after pregnancy following this model. On a policy level, it would be advisable to follow the recommendations already made by various organisations and advocate an increase in maternity leave, with special mention for self-employed women.

## Figures and Tables

**Table 1 children-09-01920-t001:** Script for breastfeeding interview.

Infant Feeding	
Expectations	Imaginary ○ Infant feeding knowledge ○ Exclusive breastfeeding opinion ○ Ideal breastfeeding
Experiences	Perceptions ○ Kind of infant feeding ○ Establishment and development of breastfeeding ○ Health problems ○ Support Feelings ○ Feel heard ○ Empowerment Something to change

**Table 2 children-09-01920-t002:** Socio-demographic characteristics of the sample population.

	Total Group (n = 22)	Rural Female (n = 11)	Urban Female (n = 11)
**Nationality**			
Spanish	19 (86.36%)	8 (72.73%)	11 (100%)
Other	3 (13.64%)	3 (27.27%)	
**Number of children**			
Mean ± SD	1.86 ± 0.71	1.54 ± 0.69	2.18 ± 0.6
1	7 (31.82%)	6 (54.55%)	1 (9.1%)
2	11 (50%)	4 (36.36%)	7 (63.64%)
3	4 (18.18%)	1 (9.1%)	3 (27.27%)

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
