# Peer review of "Perceptions of Mothers about Support and Self-Efficacy in Breastfeeding: A Qualitative Study"

_children, 2022, doi:10.3390/children9121920_

Round 1

Reviewer 1 Report

Dear authors,

I have now completed the review of the manuscript titled "Perceptions of mothers about support and self-efficacy in 2 breastfeeding: a qualitative study."

In the present study, the authors evaluated a qualitative study through semi-structured interviews.

The manuscript is interesting because it identified the beliefs and expectations of mothers concerning breastfeeding to determine the perception of their self-efficacy and the influence on the management of their babies’ feeding.

The article is well written, and I have some minor suggestions to further improve the quality of the manuscript.

1.  The background section introduced some relevant articles [12, 15–22]. Please explain the results or summarize with effect sizes. 

2. I suggest authors clarify how other researchers can obtain the original data. Where is a data availability statment?

3. Authors used MOVI-da 10! study investigating the influence on cardiovascular health and cognitive performance of a recreational and non-competitive physical activity intervention in school children. I wonder do they also have disease status. If so, it should be added in baseline characteristics.

4. What is the future scope of the proposed research, authors have described the limitations in a good way, I suggest that these can be the future scope of the work.

Author Response

Dear authors,

 I have now completed the review of the manuscript titled "Perceptions of mothers about support and self-efficacy in 2 breastfeeding: a qualitative study."

 In the present study, the authors evaluated a qualitative study through semi-structured interviews.

The manuscript is interesting because it identified the beliefs and expectations of mothers concerning breastfeeding to determine the perception of their self-efficacy and the influence on the management of their babies’ feeding.

The article is well written, and I have some minor suggestions to further improve the quality of the manuscript.

Authors: Thank you very much for your comments. We honestly think that these comments have greatly improved the manuscript.

  1. The background section introduced some relevant articles [12, 15–22]. Please explain the results or summarize with effect sizes. 

Response 1: Following the reviewer’s suggestion, we have added some information in the article.

To date, studies on breastfeeding are mainly based on the benefits of breastfeeding and those that study maternal perceptions for the most part quantify the duration of breastfeeding and associated factors which affect the breastfeeding mother, none of the qualitative studies were undertaken in Spain (12,15–22). In these studies, the researchers showed breastfeeding rates at six months, between 15.5% and 25.4%, and factors to stopping breastfeeding, such as low milk supply or work-related causes. There is a need for approaches to be undertaken as regards maternal experiences and perceptions which enables the identification of the reasons as to why mothers make decisions as regards breastfeeding and its continuance(12).

Lines: 78-80.

  1. I suggest authors clarify how other researchers can obtain the original data. Where is a data availability statment?

Response 2: In quantitative studies, it is increasingly common to provide the databases used in the analysis. However, in the case of qualitative studies, it is not a practice that is carried out. This lack of accessibility is supplied through the exact extraction of verbatims (sentences extracted from the data technique, interviews in this study) that support the themes, categories and codes analyzed. As well as the feedback given to the participants by returning the analysis of their interviews to check them. Both techniques have been performed in this study. Thus, there is not a data availability statement in our project.

  1. Authors used MOVI-da 10! study investigating the influence on cardiovascular health and cognitive performance of a recreational and non-competitive physical activity intervention in school children. I wonder do they also have disease status. If so, it should be added in baseline characteristics.

Response 3: The participants of the MOVI-da 10! study do not present any pathology or disease. Regardless of their physical condition, they became part of the study if they met the inclusion and exclusion criteria. Therefore, they are not data that we have to be able to form part of the baseline characteristics.

  1. What is the future scope of the proposed research, authors have described the limitations in a good way, I suggest that these can be the future scope of the work.

Response 4: Following the reviewer’s suggestion, we have added some information in the article.

Our findings suggest that efforts should be made at both the health and policy levels. In the former, it would be advisable to pursue breastfeeding support frameworks that go beyond the logic of the biomedical model and foster woman-centred model which takes into account the socio-cultural reality surrounding infant feeding. Thus, future research efforts should focus on undertaking interventions about breastfeeding education programs addressed to health professionals and mothers during and after pregnancy following this model. On a policy level, it would be advisable to follow the recommendations already made by various organisations and advocate an increase in maternity leave, with special mention for self-employed women.

Lines: 384-387

Reviewer 2 Report

See attached. 

Author Response

Review of: Manuscript ID: children-2076219

Title: Perceptions of mothers about support and self-efficacy in breastfeeding: a qualitative study

The study by Gálvez-Adalia and colleagues summarizes findings from a qualitative study conducted with 22 Spanish women focused on maternal perceptions of support and self-efficacy for breastfeeding. The article was easy to follow and adds to the literature on barriers and facilitators to initiation and continuation of breastfeeding. Some results described in the study agree with results from other, mostly quantitative, studies, and others disagree. I have a few minor comments for consideration.

Authors: Thank you very much for your comments. We honestly think that these comments have greatly improved the manuscript.

-Better justification on the importance of the study is needed. In lines 75-77, the authors state: To date, studies on breastfeeding are mainly based on the benefits of breastfeeding and those that study maternal perceptions for the most part quantify the duration of breastfeeding and associated factors which affect the breastfeeding mother, none of the qualitative studies were undertaken in Spain (12,15–22).” I encourage the authors to explain to the reader why might results be different in Spain and why might a qualitative study be a superior mechanism to obtain information for this particular area of research.

Response 1: Following the reviewer’s suggestion, we have added some information in the article.

To date, studies on breastfeeding are mainly based on the benefits of breastfeeding and those that study maternal perceptions for the most part quantify the duration of breastfeeding and associated factors which affect the breastfeeding mother, none of the qualitative studies were undertaken in Spain (12,15–22). In these studies, the researchers showed breastfeeding rates at six months, between 15.5% and 25.4%, and factors to stopping breastfeeding, such as low milk supply or work-related causes. There is a need for approaches to be undertaken as regards maternal experiences and perceptions which enables the identification of the reasons as to why mothers make decisions as regards breastfeeding and its continuance. Thus, understanding mothers may carry out more effective interventions to supporting and maintaining breastfeeding (12, 15-19).

Lines: 78-80 and 83-84.

-Lines 103-104 are unclear. Please rephrase.

Response 2: Following the reviewer’s suggestion, we have rewritten the sentence and added some information.

 Furthermore, the need for so-called cultural competence in healthcare personnel, mainly nurses, renders qualitative research a fundamental resource to how provide the necessary health support, or how improve the care supplied.

Lines: 107-108

-Table 1---Strange formatting. For example, some words appeared to be in bold and unclear if the dots (black and white) mean something. If they do mean something, table legend is needed.

Response 3: Following the reviewer’s suggestion, we have changed the format.

-Table 2—since no participant had 4 or more children, that category is not necessary in the table.

Response 4: Following the reviewer’s suggestion, we have removed this category.

-Briefly explain the “an/cannot conception” of breastfeeding

Response 5: Following the reviewer’s suggestion, we have added some information in order to clarify the concept.

These findings point to those obtained by Hunt et al. (36) in what is called the mechanism an/cannot conception of breastfeeding. It is defined to the author asmechanistic constructions of breastfeeding and a rules-based approach contributed towards women polarising themselves as those who could or could not breastfeed and did or did not comply with the ‘rules’ for ‘successful’ breastfeeding” (36). This mechanism is behind the discourses that place women in the group of those who have been successful in initiating and continuing breastfeeding, which are used both socially as well as medically(36–38). This mechanism is conducive to subjecting the women to Thus, women who are not able to carry out the technique of breastfeeding suffer from feelings of frustration and guilt, which may be related to depression and anxiety(38).

 Lines: 313-320

-Line 359, “heterogeneous sample.” Heterogeneous with respect to what? In terms of experience? Parity? Age? Rural/urban?

Response 6: Following the reviewer’s suggestion, we have rewritten the sentence and added some information.

Although undertaking the study in a specific setting such as the province of Cuenca may preclude the generalisation of the data, the purposive sampling strategy has ensured having a heterogeneous sample in terms of experiences and urban/rural settings. In this fashion, the representativeness of the sample has been improved so that our results can be related to the Western female population of reproductive age

Line:369

-The article has some language issues and would benefit from a review by a native English speaker.  

-For example, “teat” is usually used only for animals—authors should consider using the word “nipple” or, in some instances, “breast” would suffice.

We are changed the word teat for breast.

Response 7: Following the reviewer’s suggestion, we have changed the word teat for breast in the article.

Lines: 224,251,256,258,283.

-Rephrase line 325 “…and uncertainties as regards breastfeeding”

Response 8: Following the reviewer’s suggestion, we have rewritten the sentence.

In the first moments after birth, mothers turn to the nurses and midwives for guidance on problems and uncertainties as regards doubts regarding to breastfeeding(18).

Line: 325.

-Rephrase line 295-6 as the word “correlated” is used to denote something very specific in quantitative analysis.  

Response 9: Following the reviewer’s suggestion, we have rewritten the sentence.

The aspects which were assessed were related to their knowledge of breastfeeding, their expectations of the breastfeeding process and their experience, and their strategies for overcoming problems associated with initiating, establishing and continuing breastfeeding. These aspects were significantly correlated with the role of nurses and midwives in supporting their perception of self-efficacy.

Line: 300-301
